# Ultrafast energy relaxation dynamics of amide I vibrations coupled with protein-bound water molecules

Junjun Tan[1], Jiahui Zhang[1], Chuanzhao Li[1], Yi Luo[1] & Shuji Ye[1]

The influence of hydration water on the vibrational energy relaxation in a protein holds the key to understand ultrafast protein dynamics, but its detection is a major challenge. Here, we report measurements on the ultrafast vibrational dynamics of amide I vibrations of proteins at the lipid membrane/$H_2O$ interface using femtosecond time-resolved sum frequency generation vibrational spectroscopy. We find that the relaxation time of the amide I mode shows a very strong dependence on the $H_2O$ exposure, but not on the $D_2O$ exposure. This observation indicates that the exposure of amide I bond to $H_2O$ opens up a resonant relaxation channel and facilitates direct resonant vibrational energy transfer from the amide I mode to the $H_2O$ bending mode. The protein backbone motions can thus be energetically coupled with protein-bound water molecules. Our findings highlight the influence of $H_2O$ on the ultrafast structure dynamics of proteins.

[1] Hefei National Laboratory for Physical Sciences at the Microscale and Department of Chemical Physics, and Synergetic Innovation Center of Quantum Information & Quantum Physics, University of Science and Technology of China, 230026 Hefei, China. Correspondence and requests for materials should be addressed to Y.L. (email: yiluo@ustc.edu.cn) or to S.Y. (email: shujiye@ustc.edu.cn)

Protein–water interactions play critical roles in the structure, dynamics, and function of proteins[1–3]. Strong hydrogen bonding between proteins and water results in an intimate coupling of water thermal motion to proteins[2–4]. Although the internal motions of proteins have been proposed to be slaved to the dynamics of the surrounding water molecules in the hydration shell, few direct experimental measurements are capable of capturing these interactions on the femtosecond to picosecond timescale in which hydrogen-bonding fluctuations of water proceed[5–7]. The study of vibrational energy dissipation between proteins and their surrounding water provides a powerful approach to unravel the nature of this dynamical coupling[3,8] and analyze protein structures or solvent accessibility[9,10]. Particularly, ultrafast infrared (IR) spectroscopy such as two-dimensional (2D) IR serves as a structure-sensitive tool with fs–ps temporal resolution and has been widely used to explore the vibrational dynamics of amide I and amide A bands[11–15]. Here the amide I bands arise primarily from C=O stretching of protein backbone and are very sensitive to the hydrogen bonding between peptide bond and hydration water[16]. Because the frequency of amide I mode is strongly overlapped with the $H_2O$ bending mode

**Fig. 1** Sum frequency generation (SFG) spectra. The ssp (denoting s-, s-, and p-polarized sum-frequency output, visible input, and infrared input, respectively) SFG spectra of the peptides of amide A and amide I modes, sketch for the hydration of amide I mode, and the relationship between hydrogen–deuterium exchange (HDX) ratio and amide I frequency. **a** The ssp SFG spectra of the peptides in the N-H region. The spectra in black curves are measured at lipid bilayer/$H_2O$ interface while the ones in red curves are measured at lipid bilayer/$D_2O$ interface after 5-h HDX experiments. **b** The ssp amide I SFG spectra of peptides at lipid/$H_2O$ interface. The solid curves in **a**, **b** are the fitting curves using Supplementary Eq. 1. **c** Sketch for the carbonyl groups forming a bifurcated H bonding to water molecules upon hydration. **d** Fitting amplitude ratio $\left(\chi^{(2)}_{\text{5h HDX}}/\chi^{(2)}_{\text{before HDX}}\right)$ of the ssp SFG spectra in the N-H region after 5-h HDX and before HDX against the amide I frequency. The vertical error bars represent experimental error in determining the amplitude ratio from the fitting procedure (see Supplementary Note 2). The horizontal error bars indicate the standard error of the mean of the fitting center frequency from five independent spectra

absorption, the 2D IR experiments are generally carried out in $D_2O$ or organic solvents[11–15]. It is found that vibrational relaxation of the amide I mode occurs in ~1.2 ps (Supplementary Table 2 and Supplementary Figure 3)[11–15]. In light of this fact, the population relaxation of the amide I mode is considered as an intrinsic property of the peptide group itself and is relatively unaffected by the surrounding environment of the peptide bond and side chains[11–15]. In addition, molecular dynamics simulation of a green fluorescent protein suggested that low-frequency vibrations in proteins (collective motions of proteins) are strongly coupled with water, whereas high- and intermediate-frequency vibrations (protein backbone motions) are essentially decoupled with water[8]. It was therefore concluded that water serves as a thermal bath to enhance the intramolecular relaxation. It can be seen that, although the picture of vibrational dynamics of amide I mode in $D_2O$ and organic solvents has been extracted for the model compounds of peptide group and some globular proteins, the scenario of proteins in $H_2O$ is yet to be investigated, to the best of our knowledge. Consequently, the interaction mechanisms between proteins and $H_2O$ are not well understood and contradictory views have been reported[17–19]. Moreover, many major issues on this matter have not been addressed[3], for example, what are the roles of resonant vibrational energy transfer in proteins; is there a "shortcut" for energy to dissipate into the solvent? Because $H_2O$ is significantly different from $D_2O$ as solvents[12], the ultrafast protein dynamics in the two solvents could be quite different. Here we investigate the ultrafast vibrational dynamics of amide I mode of proteins at the $H_2O$ interface using a surface-sensitive pump–probe set-up in which a femtosecond IR pump is followed by a sum frequency generation vibrational spectroscopy (SFG-VS) probe.

SFG-VS is a powerful and versatile optical technique for characterizing the interfacial molecular structures and dynamics of various molecules including peptides and proteins in situ and in real time[20–25]. Recently, the IR pump–SFG probe technique has been employed to study the ultrafast vibrational dynamics of interfacial water and proteins by several groups[26–30]. Theoretically, the SFG signals from the water bending mode are extremely weak or not observable at all compared to amide I mode[31]. SFG-VS can therefore eliminate the interference of the $H_2O$ bending mode in the amide I region that IR spectroscopy has suffered. Indeed, many reports have confirmed that SFG spectra of amide I band are not contaminated by the contribution of $H_2O$ bending mode[32–35]. It can be naturally anticipated that the IR pump–SFG probe technique offers an effective optical method to investigate the vibrational dynamics of amide I band of interfacial proteins in $H_2O$ environments. However, to the best of our knowledge, such method has not been applied on the energy relaxation between amide I band and $H_2O$. In this study, we report measurements on the ultrafast vibrational dynamics of amide I vibrations of proteins at the lipid membrane/$H_2O$ interface using several peptides, including melittin, LKα14, mastoparan (MP), influenza M2 proteins (AM2 and BM2), and KALP23 as the models. It is known that the exposure of these peptides to water can be tuned by formation of the channel or pore in lipid membrane[36–39].

## Results

**Exposure amount of membrane-bound peptides to water**. We first determine the relative amount of peptide bonds that are exposed to $H_2O$ by measuring the ratio of the hydrogen–deuterium exchange (HDX, Supplementary Note 4) of the amide proton. Previous studies indicated that the residues exposed to $H_2O$ can undergo rapid amide proton HDX as the sample is exposed to deuterium, but the part in the core of lipid bilayer (not exposed to $H_2O$) does not exchange in several

days[30,40,41]. Figure 1a shows the amide A spectra of peptides before and after 5-h HDX. Because the interaction between peptides and membrane can cause dehydration of the membrane surface and thus suppress the SFG signals from the interfacial water molecules, the contribution from the OH signal to the signals in the amide A region is negligible[30,42]. It is evident that the ssp (denoting s-, s-, and p-polarized sum-frequency output, visible input, and infrared input, respectively) SFG spectra of melittin and LKα14 in the N-H region are almost disappeared after 5-h HDX while KALP23 changes insignificantly (Fig. 1a), illustrating that melittin and LKα14 are exposed to $H_2O$ in lipid membrane because of formation of toroidal pores, but KALP23 mainly lies in the hydrophobic core of the lipid bilayer[36]. In addition, the amide I frequency of melittin (1651 cm$^{-1}$) and LKα14 (1652 cm$^{-1}$) is observed to be lower than that of KALP23 (1665 cm$^{-1}$) (Fig. 1b). It has been reported that the amide I frequency of α-helices shifts from 1665 to 1648 cm$^{-1}$ upon hydration because the carbonyl groups in the peptide bond form a bifurcated hydrogen bonding with water molecules (Fig. 1c)[43]. Formation of a hydrogen bond with the amide oxygen can soften the vibrational potential and thereby lower the frequency[44,45]. To qualitatively determine the 5-h HDX ratio and the amide I frequency change of these peptides, we fitted the spectra using a standard procedure, Supplementary Eq. 1. We plot the fitting amplitude ratio of the ssp SFG spectra in the N-H region after 5-h HDX and before HDX ($\chi^{(2)}_{\text{5h HDX}}/\chi^{(2)}_{\text{before HDX}}$, Supplementary Table 3) against the amide I frequency (Supplementary Table 3). A linear correlation between the $\chi^{(2)}_{\text{5h HDX}}/\chi^{(2)}_{\text{before HDX}}$ ratio and the amide I frequency is clearly observed (Fig. 1d). Therefore, the amide I frequency can be used as an optical marker for the exposure amount of membrane-bound peptides to water.

**Vibrational dynamics of amide I band**. We then investigate the vibrational dynamics of amide I band at the lipid bilayer/$D_2O$ and $H_2O$ interface. Figure 2a (top panel) shows the intensity decay of the amide I band of KALP23 at the 1,2-dipalmitoyl-$sn$-glycero-3-[phospho-$rac$-(1-glycerol)] (sodium salt) (DPPG) lipid bilayer/$D_2O$ interface with $\nu_{\text{pump}} = \nu_{\text{probe}} = 1660$ cm$^{-1}$. Three-level vibrational model (Supplementary Figure 4 and Supplementary Note 5) has been used to extract the vibrational lifetime ($T_1$) and thermalization time constant ($T_{\text{th}}$) of the vibrational dynamics in many condensed matters, including water[46–48] and peptides[49–52].

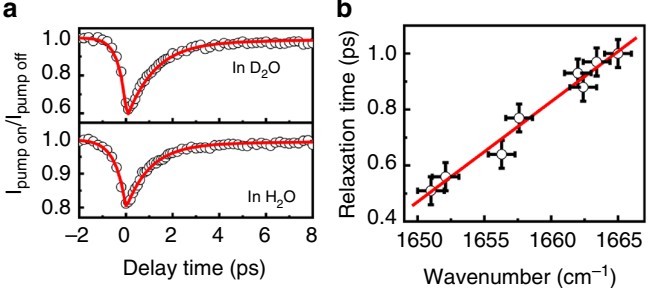

In this model, the amide I modes are excited from $v = 0$ to $v = 1$ state and it takes $T_1$ time to relax to a "hot" ground state ($v = 0^\star$), and then the intensity of the amide I band gradually recovers. The decay of the $v = 0^\star$ state with a vibrational cooling time ($T_{\text{th}}$) leads to a full thermalization of the system. The vibrational cooling time ($T_{\text{th}}$) of the amide bond has been determined to be ~10 ps[49] and is used in our fittings. It is noted that the amide I signal starts to decrease at a delay time of ~−1.2 ps in which the probe pulse significantly precedes the arrival of the pump pulse (Fig. 2a). The signal reaches a minimum at 0.1 ps and finally recovers at the positive delay time. The signal at the negative delay time is generated by the perturbed free induction decay (FID) effect[53] and can be fitted by an exponential function of $1 - A_0 \exp(t/T_0)$. FID process does not influence the data at the positive delay time[53,54]. Fit of the experimental result in Fig. 2a (top panel) for positive delay time ($t \geq 0.1$ ps) using Supplementary Eq. 16 yields a relaxation time of $T_1 = 1.15$ ($\pm 0.05$) ps, confirming the same vibrational relaxation time (~1.2 ps) discovered in $D_2O$ or organic solvents by 2D IR studies. However, at the lipid bilayer/$H_2O$ interface (Fig. 2a, bottom panel), the relaxation time is found to be 1.0 ($\pm 0.05$) ps, a little smaller than the value at the lipid bilayer/$D_2O$ interface.

**Dependence of relaxation time on $H_2O$ exposure**. Using the same experimental procedures, we investigate the vibrational dynamics of amide I band of melittin, LKα14, MP, BM2, and AM2 at the lipid membrane/$H_2O$ interface. The "pump on" and "pump off" spectra at some typical delay times and the intensity decay are shown in Supplementary Figure 5, Supplementary Figure 6, and Supplementary Note 6. Controlled experiments on the excitation of water bending mode at $CaF_2$/$H_2O$ interface and DPPG bilayer/$H_2O$ interface indicated that the effect of the excitation of water bending mode on the vibrational dynamics of amide I is very small, at least beyond the experimental errors (Supplementary Figures 7 and 8, Supplementary Note 7). The relaxation time is determined to be 0.51 ($\pm 0.05$) for melittin, 0.56 ($\pm 0.05$) for LKα14, 0.64 ($\pm 0.05$) for MP, 0.77 ($\pm 0.05$) for BM2, which are about half of the value in $D_2O$. For the AM2 samples, its relaxation time depends on the pH conditions, which are 0.88 ($\pm 0.05$) at pH = 4, 0.93($\pm 0.05$) at pH = 6.2, and 0.97($\pm 0.05$) at pH = 8.5. All these results depict that the vibrational relaxation in $H_2O$ occurs faster than that in $D_2O$. Because the peptides of LKα14, MP, KALP23, BM2, and AM2 all adopt a nearly pure α-helical structure in lipid membrane[25,55], the difference in lifetime is not contributed by the secondary structures. Summarizing all of these results, it is found that the relaxation time of amide I mode in $H_2O$ environment linearly correlates with the amide I frequency (Fig. 2b). It is worth noticing that M2 opens its channel to form a water pore at acid environment and is therefore more exposed to water[39]. The results in Fig. 2b, taking into account the relationship between the amide I frequency and the exposure amount of membrane-bound peptides to water shown in Fig. 1d, reveal that the peptides that are more exposed to $H_2O$ have a shorter relaxation time. Apparently, the relaxation time is dictated by $H_2O$ exposure, which is opposite to what has been reported for peptides in $D_2O$ or organic solvents (Supplementary Figure 3, Supplementary Note 3). The hydrogen bonding mechanism could not explain this result because $H_2O$ forms weaker hydrogen bonds than $D_2O$[56].

The relaxation of amide I mode has been assumed to be the result of intramolecular vibrational redistribution (IVR) in protein secondary structures because such fast timescale cannot be explained by direct energy transfer between the amide I mode and the surrounding solvents of the peptide group[11,13]. However, the IVR mechanism alone cannot interpret our observation that

**Fig. 2** Intensity decay of amide I band and relationship between relaxation time and amide I frequency. **a** The intensity decay of the amide I band of KALP23 at lipid membrane/water interface with $\nu_{\text{pump}} = \nu_{\text{probe}} = 1660$ cm$^{-1}$. The red lines are the fitting curves. The data for delay time at $t \leq 0.1$ ps was fitted using the function of $1 - A_0 \exp(t/T_0)$ while the data for delay time at $t \geq 0.1$ ps were fitted using Supplementary Eq. 16. **b** Relaxation time of amide I mode against the amide I frequency. The vertical error bars represent the estimated accuracy of the fit based on the time resolution of the laser pulse width. The horizontal error bars indicate the standard error of the mean of the fitting center frequency from five independent spectra

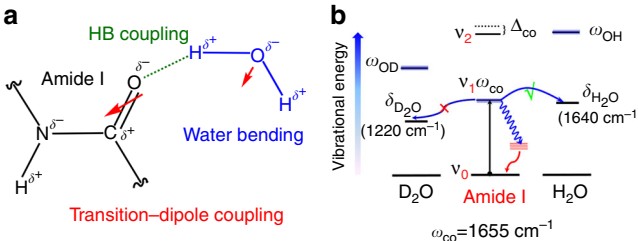

**Fig. 3** Coupling scheme and energy diagram. **a** Interaction between amide carbonyl and water through hydrogen bonding (HB) and transition–dipole coupling. **b** Energy-level depiction of the vibrational relaxation of the amide I mode to the water bending mode

$H_2O$ can induce a faster vibrational relaxation than $D_2O$, which manifests that the vibrational dynamics of amide I mode in $H_2O$ must involve other relaxation pathway through additional coupling induced by $H_2O$. The correlation between the relaxation time of amide I mode and the exposure amount of membrane-bound peptides to $H_2O$ provides experimental evidences for the transition–dipole coupling between amide I mode and the water bending mode (Fig. 3a). The vibrational mixing between them could open a resonant relaxation channel that facilitates direct vibrational energy transfer from the amide I mode to the $H_2O$ bending mode. This pathway is only present in $H_2O$ but absent in $D_2O$ because the $D_2O$ bending mode shifts to ~1220 cm$^{-1}$ and is no longer resonant with the amide I mode (Fig. 3b). Indeed, the notion of vibrational mixing in the peptide unit of $N$-methylacetamide and dipeptides has been theoretically predicted[57–59] and experimentally supported by the Raman band shape of amide I, which depends on $H_2O$ density[60–63]. In addition, such Förster-like resonant energy transfer mechanism has been observed in the case of the cyanide ion[64], cyano[65], and metal carbonyl complex[66], in which the relaxation time in $H_2O$ is faster than that in $D_2O$.

On the other hand, according to Fig. 1c, the amide I groups can actually be considered as two components: the one coupled to water and the other not coupled to water. Therefore, the intensity decay of the amide I band shown in Supplementary Figure 6 would be analyzed using two energy relaxation processes (Supplementary Note 8): one for the amide modes coupled to water with relaxation time of 0.4 ps (a value approximate to the lifetime of water bending modes[67,68]) and one for the amide modes not coupled to water with relaxation time of 1.15 ps (the lifetime of amide I mode in $D_2O$). The ratio of the component of 1.15 ps is plotted in Supplementary Figure 9. It is evident that the ratio of the component of 1.15 ps in $H_2O$ environment linearly correlates with the amide I frequency and matches well with the exposure amount of membrane-bound peptides to the water determined by HDX method (Fig. 1d). This result suggests that the relaxation time measurements combining with the site-specific labeling technique developed by Zanni et al.[35,40] will offer a unique and effective optical marker to determine the hydrophobicity of specific sites.

## Discussion
To conclude, we applied a time-resolved IR pump–SFG probe to investigate the ultrafast vibrational dynamics of amide I mode of proteins at the lipid membrane/$H_2O$ interface. We found that the relaxation of amide I mode strongly depends on the $H_2O$ exposure, in stark contrast to what was observed in the case of $D_2O$ exposure. This implies that the relaxation dynamics of amide I mode in $H_2O$ environment is not only controlled by the intrinsic property of the peptide group but also by a direct resonant

channel that is energetically coupled with protein-bound water molecules. In other words, while $D_2O$ only serves as a thermal bath to enhance the intramolecular relaxation, $H_2O$ can also provide a "shortcut" through direct resonant channel to dissipate energy into the solvent. It highlights the limitation of using $D_2O$ to study the ultrafast structure dynamics of proteins. Our findings may be important for correctly understanding the roles of resonant vibrational energy transfer through dipole coupling in protein and hydrogen bonds between $H_2O$ and the biomolecule on the entire energy transfer pathways.

## Methods
**Materials.** The peptides used in this study include melittin (a bee venom toxin, sequence: GIGAVLKVLTTGLPALISWIKRKRQQ), LKα14 (an amphiphilic peptide, sequence: LKKLLKLLKKLLKL), MP (sequence: INLKALAALAKKIL), influenza B M2 transmembrane domain (BM2, sequence: MLEPFQILSICSFIL-SALHFMAWTIGHLNQIKR), influenza A M2 transmembrane domain (AM2, sequence: SSDPLVVAASIIGILHLILWILDRL), and KALP23 (sequence: GKKLALA-LALALALALALALALKKA). These peptides have a purity of >98% and were purchased from Shanghai Apeptide Co., Ltd. The lipids of 1-palmitoyl-2-oleoyl- $sn$-glycero- 3-phospho- (1'-$rac$-glycerol) (sodium salt) (POPG), 1,2-dimyristoyl-$sn$-glycero-3-[phospho-$rac$-(1-glycerol)] (sodium salt) (DMPG), DPPG, and 1,2-dimyristoyl-d54-$sn$-glycero-3-phosphocholine-1,1,2,2-d4-N,N,N-trimethyl-d9 (d-DMPC) were purchased from Avanti Polar Lipids (Alabaster, AL). All the peptides were dissolved in methanol (purchased from Sinopharm Chemical Reagent Co., Ltd.) with different concentrations (supplementary Table 1) and stored at −20 °C. The d-DMPC phospholipid was dissolved in chloroform (purchased from Sinopharm Chemical Reagent Co., Ltd.). DPPG, DMPG, and POPG solutions (with the concentration of 1.3 mg/mL) were prepared in mixed solvents of chloroform and methanol (with a volume ratio of 2:1) (purchased from Sinopharm Chemical Reagent Co., Ltd.). All the lipid solutions were kept at −20 °C. Right-angle $CaF_2$ prisms were purchased from Chengdu Ya Si Optoelectronics Co., Ltd (Chengdu, China). All the chemicals were used as received. The cleaning of $CaF_2$ prisms and preparation of lipid bilayer were performed using a standard procedure given in Supplementary Information.

**Experiments and data analysis.** The details about SFG-VS experiments and data analysis can be found in Supplementary Information (Supplementary Note 1, Supplementary Note 2, Supplementary Note 5, Supplementary Note 6, Supplementary Figure 1, and Supplementary Figure 2).

## Data availability
The data that support the finding in current study are available from the corresponding author upon reasonable request.

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

## Acknowledgements

This work was supported by the National Key Research and Development Program of China (2017YFA0303500, 2018YFA0208700), National Natural Science Foundation of China (21873090, 21633007, 21790350), Fundamental Research Funds for the Central Universities (WK2340000064), CAS (2016HSC-IU003), and Anhui Initiative in Quantum Information Technologies (AHY090000).

## Author contributions

J.T., J.Z. and C.L. collected the SFG spectra. S.Y. processed and analyzed the data. S.Y. and Y.L. designed the study. S.Y. and Y.L. wrote the paper. All authors discussed the results of the study.

## Additional information

**Competing interests:** The authors declare no competing interests.

