## [Peer Review File · Nature Communications]

Editorial Note: Parts of this Peer Review File have been redacted as indicated as we could not obtain permission to publish the reports of reviewer #3.

Reviewers' comments:

Reviewer #1 (Remarks to the Author):

This paper describes the energy relaxation of amide I vibrations of membrane bound proteins. As the authors perform surface sensitive experiments they can measure the energy relaxation in the natural environment of H₂O. Comparable experiments in bulk can only be performed in D₂O due to overlapping of the amide I vibration with the water bending mode. The current system is thus a very good model system for natural environments.

Although the experiments are complicated, the presented data are of high quality. The authors did an excellent job in performing the experiments and in writing the paper. I am really impressed by the quality and quantity of the work. Although the results are maybe not surprising, as coupling to the bending mode might be expected, this paper is the first to demonstrate this.

I have a view comments which could improve the paper.

1. The description of the HDX experiment is very unclear. I guess the black curves in Fig. 1a are measured with H₂O in the subphase, while the red curves have for five hours D₂O in the subphase. If this is correct, than the black curves are a combination of the NH signal of the protein and the OH stretch mode of water. If the black curve is measured on D₂O, then the spectrum is not before HDX. The authors should explain in more detail in the manuscript or in the SI how the HDX experiment is performed. At least the caption of Fig. 1 should tell what the subphase is in the different experiments.

2. I am wondering how much of the time resolved signal could originate from the water bending mode. The pump will excite much more bending mode of water than amide modes, as we just have much more of it. The IR pumping is not surface specific. Could the observed pump-induced signal also be just the bending mode of water? I guess, that is unlikely, as the different systems show different relaxation times. However, another scenario could be that the bending mode is pumped and we see an effect of the heated bending mode on the amide I spectrum. The effect could be different for the different systems as the amide modes are more or less exposed to water. I would propose the authors to discuss the possible pumping of the bending mode in the manuscript.

3. Important information like the bandwidth of the pump and the time resolution (i.e. cross correlate information) is missing in the manuscript or the SI. In my opinion it would also be useful to add to the manuscript (e.g. as an additional panel in Fig. 2) or to the SI a 'pump on' divided by 'pump off' spectrum. That signal is subsequently integrated to obtain Fig. 2a. Showing the 'raw' data might be insightful also with respect to my comment 2. I would expect that if the bending mode is pumped the response is relatively broad, while for amide pumping the band should be narrower. Especially based on Fig. 1B I expect a very narrow pump-probe signal for KALP23.

4. Fig. 2B could suggest that there are two energy relaxation processes: one for the amide modes coupled to water and one for the amide modes not coupled to water. The first one has a fast relaxation time of about 400 fs, the last one might have the relaxation time of 1.2 ps like for D2O. Depending on the relative amount of each ensemble, the relaxation time is somehow a linear combination of these two timescales. What do the authors think of that? It might be instructive to add a few sentences to the manuscript regarding this interpretation.

5. Please mention in the caption of Fig. 1 what the lines in panel A and B are. Are these fits with the Lorentzian lineshape model?

6. What are the red lines in Fig. 2A and SI Fig. 3? Fits to the data? If these red lines are the fit with equation S18 described in the SI, I am wondering how the authors obtained the current curve. Apparently, they performed some convolution with the system response function and included the perturbed free induction decay. If the red lines are not the fits, it would be useful to show the fit so that the reader can judge the quality of the fit. Based on the statement "Fit of the experimental result in Figure 2A (top panel) for positive delay time..." on page 7, it seems that the red curves are not a fit as the red lines are also present at negative time.

7. For me it is not a priori clear why the relaxation of the amide I mode should occur over an intermediate level. The authors refer on page 7 to ref 25-28 to justify the model. However, 3 of these 4 references refer to the water O-H stretch mode, where it is well established that it relaxes over the bend mode to a hot ground state. I would suggest that the authors include a statement why a model with an intermediate state is justified.

8. In section 2.1 of the SI the authors write that time zero was determined by the IIV signal from IR pump, IR probe, and VIS of the amide I band. I am very surprised. In my opinion IIV is normally strong as it is a bulk signal; why should we see this from the amide I band that are solely present at the interface? Would it not be more likely that the IIV originates from the water bending mode? Please comment on this.

9. I do not agree with the last sentence of section 2.3 in the SI. The long time signal offset is caused by an increase of the temperature on ps timescale due to the heat one single laser shot deposits. This has nothing to do with the temperature of the water baths. Please correct the text.

10. The authors should mention the value of the thermalization time in the manuscript or in the SI.

Reviewer #2 (Remarks to the Author):

Tan et al. have executed a cleverly designed set of experiments to understand the vibrational energy relaxation of the amide I mode of a series of peptides. The peptides were studied in contact with a lipid membrane, and the amide I vibrational relaxation was measured with a femtosecond infrared (IR) pump pulse followed by a sum frequency generation (SFG) probe. Despite numerous previous studies in D₂O, amide I vibrational lifetimes have not been previously measured in H₂O because of the nearly resonant overlap of the H₂O bend vibration with the amide I absorption. In SFG spectroscopy, however, the water bend is significantly attenuated, and the amide I lifetimes were successfully measured. The hydration of the peptides was independently measured using H/D exchange of the amide proton. Interestingly, the amide I vibrational lifetimes correlated with the vibrational frequency and with the degree of hydration. The authors suggest that the amide I vibrational energy is dissipated directly into the bend vibrations of the solvent.

This is a scientifically sound paper, and it contains the first measurements of the amide I vibrational lifetime for peptides in water, albeit in contact with a lipid membrane. In my opinion, this paper is appropriate for a more technical journal (e.g., the Journal of Physical Chemistry Letters). The idea that amide I vibrations are resonantly coupled to H₂O bends is not new. Theoretical studies, for example, P.-A. Cazade, F. Hédin, Z.-H. Xu, and M. Meuwly, *J. Phys. Chem. B* 119, 3112-3122 (2015), have demonstrated the role of coupling to the H₂O bend on the vibrational relaxation of amide I. Although it is nice to see the theoretical predictions borne out in experiments, the prior work undermines the novelty and impact.

[Redacted]

Responses to the report of Reviewer 1:

General Comments: *This paper describes the energy relaxation of amide I vibrations of membrane bound proteins. As the authors perform surface sensitive experiments they can measure the energy relaxation in the natural environment of H₂O. Comparable experiments in bulk can only be performed in D₂O due to overlapping of the amide I vibration with the water bending mode. The current system is thus a very good model system for natural environments.*

Although the experiments are complicated, the presented data are of high quality. The authors did an excellent job in performing the experiments and in writing the paper. I am really impressed by the quality and quantity of the work. Although the results are maybe not surprising, as coupling to the bending mode might be expected, this paper is the first to demonstrate this.

Author reply: We thank the reviewer very much for his/her appreciation of our work. We are also grateful for his/her insightful comments and suggestions, which have helped us to better understand the experimental findings and to improve the presentation of our results. Our point-by-point response is given below.

I have a view comments which could improve the paper.

Comment 1): *The description of the HDX experiment is very unclear. I guess the black curves in Fig. 1a are measured with H₂O in the subphase, while the red curves have for five hours D₂O in the subphase. If this is correct, than the black curves are a combination of the NH signal of the protein and the OH stretch mode of water. If the black curve is measured on D₂O, then the spectrum is not before HDX. The authors should explain in more detail in the manuscript or in the SI how the HDX experiment is performed. At least the caption of Fig. 1 should tell what the subphase is in the different experiments.*

Author reply 1): The spectra in black curves of Fig. 1a are measured at lipid bilayer/H₂O interface while the ones in red curves are measured at lipid bilayer/D₂O interface after 5-h HDX experiment.

To address this comment, we have revised “**Figure 1. A)** The ssp SFG spectra of the peptides in the NH region, the spectra before HDX in black and the one after 5h HDX in red”

into “**Figure 1. A)** The ssp SFG spectra of the peptides in the NH region. The spectra in black curves are measured at lipid bilayer/H₂O interface while the ones in red curves are measured at lipid bilayer/D₂O interface after 5-h HDX experiments” (Page 7).

We have also added a paragraph in Supplementary Information (Page S6) to describe the experimental procedure, which reads: “We first prepared the membrane-bound peptides at the subphase of H₂O. After the interactions between peptides and lipid bilayers reach equilibrium, we measured the SFG spectra in the amide A band and amide I bands. And then we replaced the subphase solution of lipid bilayer by D₂O carefully. After 5 h to allow HDX to take place, we collected SFG spectra of the peptides at lipid bilayer/D₂O interface again. Figure 1A shows the ssp SFG spectra of the amide A band at lipid membrane interface. The spectra in black curves are measured at lipid bilayer/H₂O interface while the one in red curved are measured at lipid bilayer/D₂O interface after 5-h HDX experiments”.

Comment 2): *I am wondering how much of the time resolved signal could originate from the water bending mode. The pump will excite much more bending mode of water than amide modes, as we just have much more of it. The IR pumping is not surface specific. Could the observed pump-induced signal also be just the bending mode of water? I guess, that is unlikely, as the different systems show different relaxation times. However, another scenario could be that the bending mode is pumped and we see an effect of the heated bending mode on the amide I spectrum. The effect could be different for the different systems as the amide modes are more or less exposed to water. I would propose the authors to discuss the possible pumping of the bending mode in the manuscript.*

Author reply 2): We thank the reviewer for this excellent suggestion. In order to reduce the effect of the excitation of water bending mode on the amide I modes, we used a home-made water bath (see Supplementary Figure 1) to maintain the sample temperature at 24°C. The subphase of the bilayer was surrounded by the flowing water in water bath, which can effectively avoid the heating caused by the laser. In addition, we also performed controlled experiments to investigate the influence of the excitation of water bending mode on the amide I modes. It is found that the bleaching in frequency range of 1600-1700 cm⁻¹ is negligible when we pump the water bending mode at CaF₂/H₂O interface and DPPG bilayer/ H₂O

interface. Therefore, the effect of the excitation of water bending mode on the vibrational dynamics of amide I is very small, at least beyond the experimental errors.

To address this comment, we have added following contents in Supplementary Information (Page S13-S15)

8. Controlled experiments on the excitation of water bending mode at CaF₂/H₂O interface and DPPG bilayer/ H₂O interface

We have performed controlled experiments to investigate the influence of the excitation of water bending mode on the amide I modes. It is found that the bleaching in frequency range of 1600-1700 cm⁻¹ is negligible when we pump the water bending mode at CaF₂/H₂O interface and DPPG bilayer/ H₂O interface. Therefore, the effect of the excitation of water bending mode on the vibrational dynamics of amide I is very small, at least beyond the experimental errors.

Figure S7. A) The static ssp SFG spectrum at CaF₂/water interface: the red solid dot represents the ‘pump on’ spectrum at delay time of 0.2 ps, the black dot represents the ‘pump off’ spectrum, the blue triangle is the corresponding differential transient SFG spectra ($I_{\text{pump on}} - I_{\text{pump off}}$). B) The intensity decay of water bending mode with $\nu_{\text{pump}} = 1660\text{cm}^{-1}$ and $\nu_{\text{probe}} = 1660\text{cm}^{-1}$. C) The differential transient SFG spectra ($I_{\text{pump on}} - I_{\text{pump off}}$) in the region of water bending mode. To avoid the spectra to overlap together, the spectra were offset with a certain value.

Figure S8. A) The static ssp SFG spectrum at CaF₂-supporting DPPG bilayer/water interface: the red solid dot represents the ‘pump on’ spectrum at delay time of 0.2ps, the black dot represents the ‘pump off’ spectrum, the blue triangle is the corresponding differential transient SFG spectra ($I_{\text{pump on}} - I_{\text{pump off}}$). B) The intensity decay of water bending mode with $\nu_{\text{pump}} = 1660 \text{ cm}^{-1}$ and $\nu_{\text{probe}} = 1660 \text{ cm}^{-1}$. C) The differential transient SFG spectra ($I_{\text{pump on}} - I_{\text{pump off}}$) in the region of water bending mode. To avoid the spectra to overlap together, the spectra were offset with a certain value.

Comment 3): *Important information like the bandwidth of the pump and the time resolution (i.e. cross correlate information) is missing in the manuscript or the SI. In my opinion it would also be useful to add to the manuscript (e.g. as an additional panel in Fig. 2) or to the SI a ‘pump on’ divided by ‘pump off’ spectrum. That signal is subsequently integrated to obtain Fig. 2a. Showing the ‘raw’ data might be insightful also with respect to my comment 2. I would expect that if the bending mode is pumped the response is relatively broad, while for amide pumping the band should be narrower. Especially based on Fig. 1B I expect a very narrow pump-probe signal for KALP23.*

Author reply 3): Following the valuable suggestion of the reviewer, we have made following revisions in Supplementary Information.

- 1) We have added “In this study, the bandwidth of the pump IR and probe IR pulses is about 170 cm^{-1} (FWHM)¹⁰”(Page S2)

2) We have added following contents in Supplementary Information: The IIV-SFG cross-correlation traces of water bending mode at peptide-inserted DPPG bilayers /water interface are given in Supplementary Figure 2. The full width at half maximum is about 285 ± 5 fs. According to the coherence length, the IIV-SFG only probes the bulk molecules in the interfacial distance below ~ 80 nm. Therefore, the IIV SFG signals from the water bending mode can be used to determine the time-zero and instrument response of the IR pump- SFG probe process (Page S2-S3)

Supplementary Figure 2. The IIV-SFG cross-correlation traces of water bending mode at different peptide-inserted DPPG bilayer/water interface under current experimental geometry. A) pure DPPG bilayer/water; B) KALP23- inserted DPPG bilayer/water; C) melittin-inserted DPPG bilayer/water.

6. The ‘pump on’ and ‘pump off’ spectra at some typical delay times(Page S10-S12)

Supplementary Figure 5. The ‘pump off’ (red curve) and ‘pump on’ (black curve) spectra at different delay times. a) Melittin; b) LK α 14; c) MP; d) BM2; e) AM2 at pH =4; f) AM2 at pH =6.2; g) AM2 at pH =8.5; h) KALP23 (H₂O); i) KALP23 (D₂O).

Supplementary Figure 5 shows the ‘pump off’ (red curve) and ‘pump on’ (black curve) spectra at different delay times. It can be seen that the bandwidth of the ‘pump on’ spectra of amide I modes at different delay times is similar to the one of the ‘pump off’ spectra. In general, if the excitation of the water bending mode affects the dynamics of amide I mode, the bandwidth will become broader. Therefore, the influence of the excitation of water bending mode can be excluded. The intensity ratio of $I_{\text{pump on}}/I_{\text{pump off}}$ is obtained (Supplementary Figure 6) by integrating the spectra in the frequency from $\omega_v - \Gamma_v$ to $\omega_v + \Gamma_v$ at each delay time. Here, the ω_v and Γ_v are peak center and damping coefficient of the Amide I mode given by Supplementary Eq. S1, respectively. For melittin, the intensity ratio of $I_{\text{pump on}}/I_{\text{pump off}}$ is obtained by integrating the spectra in the frequency from 1640 cm⁻¹ to 1670 cm⁻¹ at each delay time.

Comment 4): *Fig. 2B could suggest that there are two energy relaxation processes: one for the amide modes coupled to water and one for the amide modes not coupled to water. The*

first one has a fast relaxation time of about 400 fs, the last one might have the relaxation time of 1.2 ps like for D₂O. Depending on the relative amount of each ensemble, the relaxation time is somehow a linear combination of these two timescales. What do the authors think of that? It might be instructive to add a few sentences to the manuscript regarding this interpretation.

Author reply 4): Following the valuable suggestion of the reviewer, we have analyzed the intensity decay of the amide I band shown in Supplementary Figure 6 using two energy relaxation processes: one for the amide modes coupled to water with relaxation time of 0.4 ps (a value approximate to the lifetime of water bending modes) and one for the amide modes not coupled to water with relaxation time of 1.15 ps (the lifetime of amide I mode in D₂O). The ratio of the component of 1.15 ps is plotted in Supplementary Figure 9. It is evident that the ratio of the component of 1.15 ps in H₂O environment linearly correlates with the amide I frequency and matches well with the exposure amount of membrane-bound peptides to the water determined by HDX method (Figure 1D). This result suggests that the relaxation time measurements combining with the site-specific labeling technique developed by Zanni et al. will offer a unique and effective optical marker to determine the hydrophobicity of specific sites.

To address this comment, we have added “On the other hand, according to Figure 1A, the amide I groups can actually be considered as two components: the one coupled to water and the other not coupled to water. Therefore, the intensity decay of the amide I band shown in Supplementary Figure 6 would be analyzed using two energy relaxation processes: one for the amide modes coupled to water with relaxation time of 0.4 ps (a value approximate to the lifetime of water bending modes^{66,67}) and one for the amide modes not coupled to water with relaxation time of 1.15 ps (the lifetime of amide I mode in D₂O). The ratio of the component of 1.15 ps is plotted in Supplementary Figure 9. It is evident that the ratio of the component of 1.15 ps in H₂O environment linearly correlates with the amide I frequency and matches well with the exposure amount of membrane-bound peptides to the water determined by HDX method (Figure 1D). This result suggests that the relaxation time measurements combining with the site-specific labeling technique developed by Zanni et al.^{35,40} will offer a unique and effective optical marker to determine the hydrophobicity of specific sites” (Page 11).

We have added following contents in Supplementary Information (Page S15)

9. Analyzing using two energy relaxation processes

According to Figure 1A, the amide I groups can actually be considered as two components: the one coupled to water and the other not coupled to water. Therefore, the intensity decay of the amide I band shown in Supplementary Figure 6 could also be analyzed using two energy relaxation processes: one for the amide modes coupled to water with relaxation time of 0.4 ps (a value approximate to the lifetime of water bending modes^{32,33}) and one for the amide modes not coupled to water with relaxation time of 1.15 ps (the lifetime of amide I mode in D₂O). The ratio of the component of 1.15 ps is plotted in Supplementary Figure 9. It is evident that the ratio of the component of 1.15 ps in H₂O environment linearly correlates with the amide I frequency and matches well with the exposure amount of membrane-bound peptides to the water determined by HDX method (Figure 1D). This result suggests that the relaxation time measurements combining with the site-specific labeling technique developed by Zanni et al.^{34,35} will offer a unique and effective optical marker to determine the hydrophobicity of specific sites.

Supplementary Figure 9. Red: the ratio of the component of 1.15 ps component is plotted against the amide I frequency; Black: fitting amplitude ratio ($\chi_{5h\ HDX}^{(2)} / \chi_{before\ HDX}^{(2)}$) of the ssp SFG spectra in the N-H region after 5h HDX and before HDX against the amide I frequency.

Comment 5): Please mention in the caption of Fig. 1 what the lines in panel A and B are. Are this fits with the lorentzian lineshape model?

Author reply 5): To address this comment, we have added “The solid curves in Figure 1A and 1B are the fitting curves using Supplementary Eq. S1” (Page 7)

Comment 6): *What are the red lines in Fig. 2A and SI Fig. 3? Fits to the data? If these red lines are the fit with equation S18 described in the SI, I am wondering how the authors obtained the current curve. Apparently, they performed some convolution with the system response function and included the perturbed free induction decay. If the red lines are not the fits, it would be useful to show the fit so that the reader can judge the quality of the fit. Based on the statement “Fit of the experimental result in Figure 2A (top panel) for positive delay time....” on page 7, it seems that the red curves are not a fit as the red lines is also present at negative time.*

Author reply 6): The red lines in Fig. 2A and SI Fig.6 are the fitting curves. We fitted the data for delay time at $t \leq 0.1$ ps using the function of $1-A_0 \exp(t/T_0)$ and fitted the data for delay time at $t \geq 0.1$ ps using Supplementary Eq. S18 for the peptides of Melittin, LK α 14, MP and AM2. For the peptides of BM2 and KALP23, Supplementary Eq. S16 was used because of relatively large bleaching.

To address this comment, we have added: “The red lines are the fitting curves. The data for delay time at $t \leq 0.1$ ps was fitted using the function of $1-A_0 \exp(t/T_0)$ while the data for delay time at $t \geq 0.1$ ps were fitted using Supplementary Eq. S16” in main text (Page 9) and “The red lines are the fitting curves. We fitted the data for delay time at $t \leq 0.1$ ps using the function of $1-A_0 \exp(t/T_0)$ and fitted the data for delay time at of $t \geq 0.1$ ps using Supplementary Eq. S18 for the peptides of Melittin, LK α 14, MP and AM2. For the peptides of BM2 and KALP23, the data for delay time at $t \geq 0.1$ ps was fitted using Supplementary Eq. S16 because of relatively large bleaching” in Supplementary Information (Page S13).

Comment 7): *For me it is not a priori clear why the relaxation of the amide I mode should occur over an intermediate level. The authors refer on page 7 to ref 25-28 to justify the model. However, 3 of these 4 references refer to the water O-H stretch mode, where it is well established that it relaxes over the bend mode to a hot ground state. I would suggest that the authors include a statement why a model with an intermediate state is justified.*

Author reply 7): We thank the reviewer for the good suggestion. We have re-analyzed the relaxation dynamics using three-level model (Supplementary Figure 4) (Page S6-S9).

To address this comment, we have replaced the sentence “Following the excitation of amide I mode from the ground ($v=0$) to its first excited vibrational state ($v=1$), the SFG intensity decrease. As the $v=1$ state relaxes to an intermediate state (v^*) and then to the “hot” ground state (v^{**})²⁵⁻²⁸, the intensity of the amide I band gradually recovers” by a discussion of a new model, i.e. “Three-level vibrational model (**Supplementary Figure 4**) has been used to extract the vibrational lifetime (T_1) and thermalization time constant (T_{th}) of the vibrational dynamics in many condensed matters, including water⁴⁵⁻⁴⁷ and peptides⁴⁸⁻⁵¹. In this model, the amide I modes are excited from $v=0$ to $v=1$ state and it takes T_1 time to relax to a “hot” ground state ($v=0^*$), and then the intensity of the amide I band gradually recovers. The decay of the $v=0^*$ state with a vibrational cooling time (T_{th}) leads to a full thermalization of the system. The vibrational cooling time (T_{th}) of the amide bond has been determined to be ~ 10 ps⁴⁸ and was used in our fittings” (**Page 7-8**).

It is worth mentioning that according to the equations describing the intensity decay for the three-level and four-level vibrational models (Eqs.R1 and R2), the influence of different models on the vibrational lifetime (T_1) of amide I mode is very small because of the small contribution of the vibrational cooling part (less than 10% in our study).

Three-level model:

$$\frac{I_{pump\ on}}{I_{pump\ off}} = 1 - 2(1 - \sqrt{S_0} - \Delta S)e^{-\frac{t-t_0}{T_1}} - 2\Delta S e^{-\frac{t-t_0}{T_{th}}} + (1 - \sqrt{S_0} - \Delta S)^2 e^{-2(t-t_0)/T_1} \quad (R1)$$

Four-level model:

$$\frac{I_{pump\ on}}{I_{pump\ off}} = 1 - 2(1 - \sqrt{S_0})e^{-(t-t_0)/T_1} + 2(\sqrt{1 + \Delta S} - 1)(1 - e^{-(t-t_0)/T_{th}}) + (1 - \sqrt{S_0})^2 e^{-2(t-t_0)/T_1} \quad (R2)$$

Comment 8): *In section 2.1 of the SI the authors write that time zero was determined by the IIV signal from IR pump, IR probe, and VIS of the amide I band. I am very surprised. In my opinion IIV is normally strong as it is a bulk signal; why should we see this from the amide I band that are solely present at the interface? Would it not be more likely that the IIV originates from the water bending mode? Please comment on this.*

Author reply 8): The reviewer is right. The IIV signals originate from the water bending

mode. According to this comment, we have revised “by monitoring the third-order cross-correlation of infrared-infrared visible(IIV) sum-frequency signals from the amide I bands of the peptides as a function of the delay between the IR pump and IR probe pulses” into “by monitoring the third-order cross-correlation of infrared-infrared visible (IIV) sum-frequency signals from the water bending mode as a function of the delay between the IR pump and IR probe pulses. The IIV-SFG cross-correlation traces of water bending mode at peptide-inserted DPPG bilayers/ water interface are given in Supplementary Figure 2. The full width at half maximum is about 285 ± 5 fs. According to the coherence length, the IIV-SFG only probes the bulk molecules in the interfacial distance below ~ 80 nm. Therefore, the IIV SFG signals from the water bending mode can be used to determine the time-zero and instrument response of the IR pump- SFG probe process”(Page S2-S3)

Comment 9): *I do not agree with the last sentence of section 2.3 in the SI. The long time signal offset is caused by an increase of the temperature on ps timescale due to the heat one single laser shot deposits. This has nothing to do with the temperature of the water baths. Please correct the text.*

Author reply 9): Following the valuable suggestion of the reviewer, we have deleted “as indicated by the tiny long time signal offsets” (Page S4).

Comment 10): *The authors should mention the value of the thermalization time in the manuscript or in the SI*

Author reply 10): Following the valuable suggestion of the reviewer, we have added “The vibrational cooling time (T_{th}) of the amide bond has been determined to be ~ 10 ps⁴⁸ and was used in our fittings” in the main text (Page 8).

Responses to the report of Reviewer 2:

General Comments: *Tan et al. have executed a cleverly designed set of experiments to understand the vibrational energy relaxation of the amide I mode of a series of peptides. The peptides were studied in contact with a lipid membrane, and the amide I vibrational relaxation was measured with a femtosecond infrared (IR) pump pulse followed by a sum frequency generation (SFG) probe. Despite numerous previous studies in D₂O, amide I vibrational lifetimes have not been previously measured in H₂O because of the nearly resonant overlap of the H₂O bend vibration with the amide I absorption. In SFG spectroscopy, however, the water bend is significantly attenuated, and the amide I lifetimes were successfully measured. The hydration of the peptides was independently measured using H/D exchange of the amide proton. Interestingly, the amide I vibrational lifetimes correlated with the vibrational frequency and with the degree of hydration. The authors suggest that the amide I vibrational energy is dissipated directly into the bend vibrations of the solvent.*

This is a scientifically sound paper, and it contains the first measurements of the amide I vibrational lifetime for peptides in water, albeit in contact with a lipid membrane. In my opinion, this paper is appropriate for a more technical journal (e.g., the Journal of Physical Chemistry Letters). The idea that amide I vibrations are resonantly coupled to H₂O bends is not new. Theoretical studies, for example, P.-A. Cazade, F. Hédin, Z.-H. Xu, and M. Meuwly, J. Phys. Chem. B 119, 3112-3122 (2015), have demonstrated the role of coupling to the H₂O bend on the vibrational relaxation of amide I. Although it is nice to see the theoretical predictions borne out in experiments, the prior work undermines the novelty and impact.

Author reply: We thank the reviewer for his/her appreciation of our experiments and for stating that “**This is a scientifically sound paper, and it contains the first measurements of the amide I vibrational lifetime for peptides in water**”.

It is true that theoretical studies on the peptide unit of N-methylacetamide and dipeptides have predicted the role of coupling to the H₂O bending on the vibrational relaxation of amide I. However, such prediction has not been verified by any experiments, and its correctness has not been examined for large peptide or protein systems. Even more, molecular dynamics simulations of a green fluorescent protein suggested that low-frequency vibrations in proteins (collective motions of proteins) are strongly coupled with water while high- and

intermediate-frequency vibrations (protein backbone motions) are essentially decoupled with water. In other words, how the amide I mode coupled with H₂O in real proteins is an important issue that needs to be investigated, while the lack of effective detection for the amide I groups at the H₂O interface (one of the most nature and important scenarios) with fs-ps temporal resolution has made it impossible to be experimentally verified.

In this paper, we applied time-resolved femtosecond SFG technique and have successfully detected the ultrafast vibrational energy relaxation involving amide I vibrational mode in the proteins at the cell membrane/H₂O interface for the first time. We have obtained at least two fundamentally important findings from this study. One of them is the first determination of the vibrational energy transfer time of amide I mode in H₂O environment. Moreover, it is found that the relaxation time shows a very strong dependence on the H₂O exposure, but not on D₂O exposure. The second finding is related to coupling of protein backbone motion with protein-bound water molecules. It is evident that the relaxation dynamics of amide I mode in H₂O environment is not only controlled by the intrinsic property of the peptide group, but also by a direct resonant channel that is energetically coupled with protein-bound water molecules. In other words, while D₂O only serves as a thermal bath to enhance the intramolecular relaxation, H₂O can also provide a “shortcut” through direct resonant channel to dissipate energy into the solvent. This study highlights the limitation of using D₂O to study the structure dynamics of proteins, which is a strategy that has been frequently employed in many studies. In addition, the relaxation time measurements combining with the site-specific labeling technique will offer a unique and effective optical marker to determine the hydrophobicity of specific sites.

We would like to point out that all of three reviewers agreed that our study **reports the first measurements of the amide I vibrational lifetime for peptides in water and demonstrates the coupling between amide I modes of large peptides and water bending mode.**

To address this comment, we have revised “the notion of vibrational mixing is also supported by the Raman band shape of amide I in which depends on H₂O density⁴⁷” into “the notion of vibrational mixing in the peptide unit of N-methylacetamide(NMA) and dipeptides

has been theoretically predicted⁵⁶⁻⁵⁸ and experimentally supported by the Raman band shape of amide I in which depends on H₂O density⁵⁹⁻⁶², (Page 11).

[Redacted]

REVIEWERS' COMMENTS:

Reviewer #1 (Remarks to the Author):

The authors have greatly improved the paper based on the comments of the three reviewers. As such I recommend publication in Nature Communication.

One last remark: In the revised version of the paper it is now clear to me what is plotted in Figure 1A. However, I am very puzzled how this experiment works. The black curve is measured on water and should thus contain a OH signal from water and a NH signal from the peptide. The red signal only contains the NH signal as the experiment is performed on D₂O. How could the authors conclude in panel D of this figure that they measure solely the NH vibration? Is the OH signal from water so small that it is negligible?

[Redacted]

Responses to the report of Reviewer 1:

General Comments: *The authors have greatly improved the paper based on the comments of the three reviewers. As such I recommend publication in Nature Communication.*

One last remark: In the revised version of the paper it is now clear to me what is plotted in Figure 1A. However, I am very puzzled how this experiment works. The black curve is measured on water and should thus contain a OH signal from water and a NH signal from the peptide. The red signal only contains the NH signal as the experiment is performed on D2O. How could the authors conclude in panel D of this figure that they measure solely the NH vibration? Is the OH signal from water so small that it is negligible?

Author reply: In our study, the OH signal from the peptide-bound membrane surface is negligible because the interaction between peptides and membrane can cause dehydration of the surface and thus suppress the SFG signals from the interfacial water molecules, which has been improved by previous SFG studies (see references: *Chin. J. Chem. Phys.* **31**, 523-528 (2018); *Angew. Chem. Int. Ed.* **56**, 12977-12981 (2017)).

To address this comment, we have added “Because the interaction between peptides and membrane can cause dehydration of the membrane surface and thus suppress the SFG signals from the interfacial water molecules, the contribution from the OH signal to the signals in the amide A region is negligible^{30, 42}.” (Page 5)

[Redacted]